# Branch-level Network Re-parameterization with Neural Substitution

## Abstract

In this paper, we propose the neural substitution method for network re-parameterization at branch-level connectivity. The proposed neural substitution method learns a variety of network topologies, allowing our re-parameterization method to exploit the ensemble effect fully. In addition, we introduce a guiding method for reducing the non-linear activation function in a linear transformation. Because branch-level connectivity necessitates multiple non-linear activation functions, they must be infused to a single activation with our guided activation method during the re-parameterization process. Being able to reduce the non-linear activation function in this manner is significant as it overcomes the limitation of the existing re-parameterization method, which works only at block-level connectivity. If re-parameterization is applied only at the block-level connectivity, the network topology can only be exploited in a limited way, which makes it harder to learn diverse feature representations. On the other hand, the proposed approach learns a considerably richer representation than existing re-parameterization methods due to the unlimited topology, with branch-level connectivity, providing a generalized framework to be applied with other methods. The proposed method improves the re-parameterization performance, but it is also a general framework that enables existing methods to benefit from branch-level connectivity. In our experiments, we show that the proposed re-parameterization method works favorably against existing methods while significantly improving their performance. Upon acceptance, we will make our implementation publicly available.

## 1 Introduction

Heavy architectural design choice employing multi-branch block has been thought to be a viable strategy for improving the performance of deep neural networks; however, these strategies have the downside of greatly increasing computation time. As a result, re-parameterization (Ding et al., 2021c; 2019; 2021b; Chen et al., 2019; Guo et al., 2020; Cao et al., 2022; Huang et al., 2022; Zhang et al., 2021; Ding et al., 2021a) approaches have been investigated in order to solve these computational complexity problems while retaining the benefits of the heavy multi-branch design by reducing them to a single branch.

RepVGG (Ding et al., 2021c) was the first attempt to introduce re-parameterization, which separately updates the weights of each branch during training and merges the weight of multiple convolution filter branches into a single one during inference. Many subsequent studies (Ding et al., 2019; Zhang et al., 2021; Liu et al., 2018; Ding et al., 2021a; Huang et al., 2022) exploited more diversified kernels and inception-style block designs, DBB (Ding et al., 2021b), to improve the re-parameterization method. These studies are block-level re-parameterization strategies that use multiple convolution branches within a single block. Due to the block-level connectivity that forwards the aggregated multiple outputs to the next block, as shown in Figure. 1a, it is prohibited to connect at the branch-level as illustrated in Figure. 1b.

We point out as a shortcoming that existing approaches do not benefit of generating richer information likes branch-level connectivity. We argue that the branch-level connectivity allows each layer to learn a robust information flow. The reason for this is that branch-level connectivity enables the creation of a nearly unlimited number of sub-network topologies within a network, enhancing the potential advantages of ensembles. Furthermore, learning different sub-network topologies serves as a kind

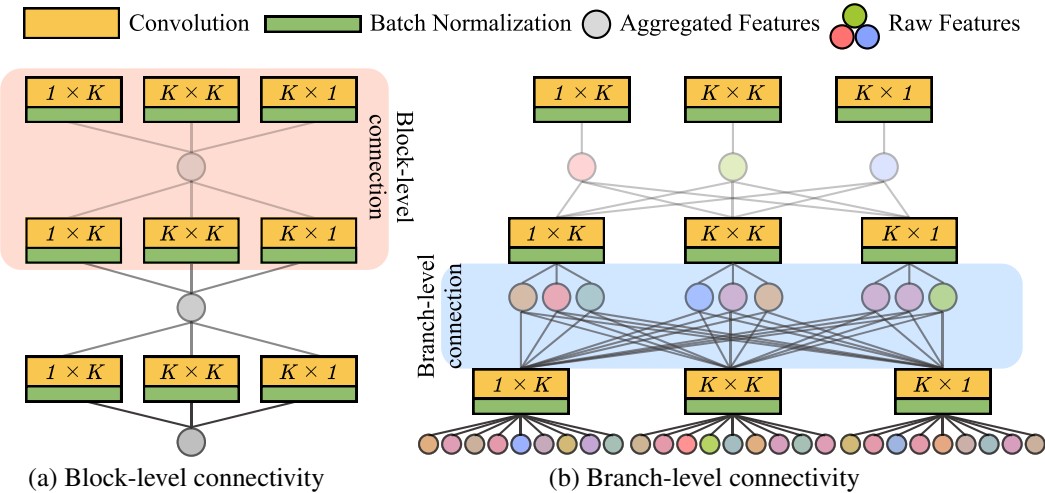

Figure 1: Progression of block- and branch-level connectivity in a re-parameterization network. (a) Block-level aggregates all branch outputs into a single feature, but (b) branch-level generates amplified output features according to the number of branches.

of regularization method, facilitating a network to profit from more performance improvement as the number of branches in each block increases. Based on these considerations, a network should be forwarded with the branch-level connectivity; nonetheless, there are two essential difficulties that must be addressed. The first is that as the network's depth increases, so does the number of topologies that must be computed. The last layer of a network with multi-branches will have near-infinite topology. This leads to an overly complex network design that cannot be employed in practice. The second difficulty is to infuse the non-linear activation functions of each branch in the re-parameterization procedure. The linear convolution and batch normalization layer can be reduced by a linear transformation in existing re-parameterization methods; however, non-linear activation functions are not achievable. This means even though the network expands, it doesn't increase non-linearity. We address these two difficulties in branch-level re-parameterizable networks by proposing 1) neural substitution and 2) guided activation method. The proposed method substitutes multiple input features with reduced numbers in order to substantially lower the computational cost. Further, using our guided activation method, the non-linear activation function can also be infused by the linear transformation in the re-parameterization procedure. Therefore, the proposed substitution with a guided activation method retains the benefits of the branch-level re-parameterizable networks without heavy architectural design and performs favorably compared to the existing re-parameterization methods. We designate this substitution-based approach for re-parameterization as the Neural Substitution (NS).

The contribution of our paper is summarized as follows:

- We introduce the neural substitution for network re-parameterization at branch-level connectivity. In the network training process, we can preserve the branch-level connectivity without being too heavy in architectural design by substituting local paths.

- We present a strategy for reducing the non-linear activation function in linear transformation during the re-parameterization process. We allow the activated neuron of each branch to be determined by a guiding method so that the non-linear activation function can be infused.

- We propose a generalized framework that can be applied to most network designs with branch-level connectivity and re-parameterizable non-linear activation functions. We demonstrate that the proposed method performs well compared to existing re-parameterization methods and that they can be adapted to our framework.

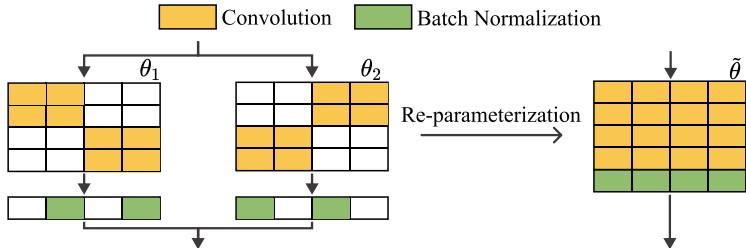

Figure 2: Basic principle of the re-parameterization. When each branch of the convolution and batch normalization layer has its unique parameters, $\theta_1$ and $\theta_2$ can be reduced by $\tilde{\theta}$ through the linear transformation.

## 2 BACKGROUND

### 2.1 OVERVIEW OF RECENT RE-PARAMETERIZATION METHOD

RepVGG (Ding et al., 2021c) was the first attempt to introduce the structural re-parameterization method. It produces multiple $3 \times 3$ convolution branches, which are re-parameterized into a single $3 \times 3$ convolution, which maintains the efficacy of the multiple branches. Inspired by the success of RepVGG, ACNet (Ding et al., 2019) proposed the re-parameterization method based on the diversified branches with different types of convolution branches. Subsequently, the Inception-style (Szegedy et al., 2015) branches, large kernel convolution, and neural architecture search (Zoph & Le, 2016) are applied to DBB (Ding et al., 2021b), MobileOne (Vasu et al., 2023), RepLKNet (Ding et al., 2022), and RepNas (Zhang et al., 2021). Recently, there have been efforts to reduce the computational complexity of re-parameterization (Hu et al., 2022; Huang et al., 2022).

Based on its use, re-parameterization is classified into two approaches. First, parameter re-parameterization (Liu et al., 2018; Cao et al., 2022; Chen et al., 2019; Guo et al., 2020) is an approach for transferring a meta learner to parameters of a new model, such as NAS or knowledge distillation methods. The second, which is the subject of this paper, is structural re-parameterization (Zhang et al., 2021; Huang et al., 2022; Ding et al., 2021b; 2019; 2021c) approach replacing the parameters of multiple convolution and batch normalization (Ioffe & Szegedy, 2015) layers to new one via a linear transformation, as illustrated in Figure. 2. To demonstrate this structural re-parameterization, we must first describe the operations of the convolution and batch-normalization layers as follows:

$$\mathcal{F}(x;\theta) = \frac{\gamma}{\sigma}(x * k + \mu) + \beta, \tag{1}$$

where $x$ denotes the input feature and $\theta = (k, \gamma, \sigma, \mu, \beta)$ stands for the set of parameters of convolution and batch normalization layers. Because the function $\mathcal{F}$ is made up entirely of linear functions, parameters of multiple convolution and batch normalization layers are reducible by following linear transformation $T(\cdot)$:

$$T\left(\sum_{i=1}^{N} \theta_i\right) = T(\theta_1) + T(\theta_2) + ... + T(\theta_N)$$

$$\text{s.t.} \quad \mathcal{F}(x;\tilde{\theta}) = \mathcal{F}(x;\theta_1) + \mathcal{F}(x;\theta_2) ... + \mathcal{F}(x;\theta_N).$$

This reduction to a single $\tilde{\theta}$ parameter set serves as the basis for the (structural) re-parameterization method. However, this raises a critical issue that the non-linear activation function can not be used in the re-parameterization method as:

$$\sigma(\mathcal{F}(x;\theta_1)) + \sigma(\mathcal{F}(x;\theta_2)) \neq \mathcal{F}(x;\tilde{\theta}). \tag{2}$$

If the network fails to become non-linear, ultimately, increasing the number of learnable parameters is not hold significant meaning. We will discuss in more detail in Sec. 4.2 whether there is a way to add non-linearity in re-parameterizable convolution blocks.

## 3 VIEW POINT OF CONNECTIVITY.

In this section, we compare the block- and branch-level connectivity regarding to leraning the feature diversity from the viewpoint of a connectivity methods.

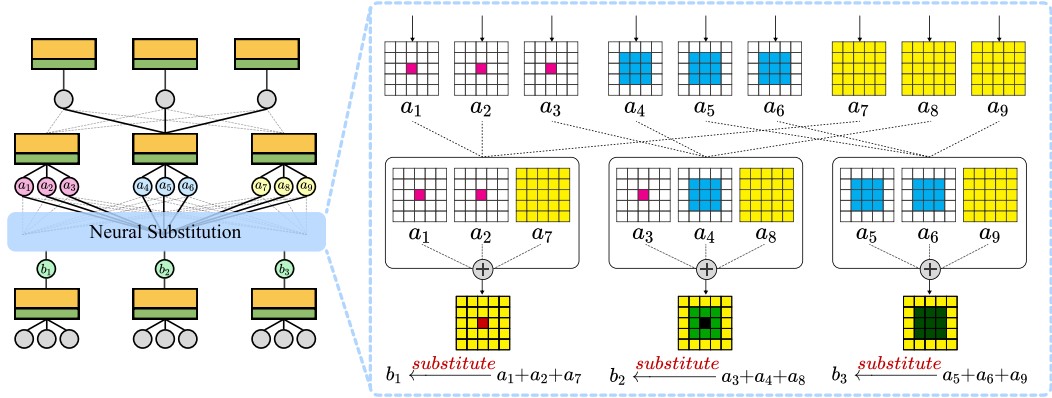

Figure 3: The conceptual visualization of the stochastic neural substitution. Raw features $a_1, ..., a_9$ in a block is substituted by $b_1, b_2, b_3$ to reduce the computational complexity of the following blocks.

## 3.1 BLOCK-LEVEL CONNECTIVITY

Many models (Ding et al., 2021b; Szegedy et al., 2015; Ding et al., 2019; 2021c; Gao et al., 2019) have employed the concept of using multiple branches in parallel within a single block design. These models add or concatenate feature outputs from each branch and feed them into the following block as follows:

$$y = \sum_{i=1}^{N} \mathcal{F}(x, \theta_i); \quad \Theta = \{\theta_1, \theta_2, ..., \theta_N\}, \tag{3}$$

where $N$ is the number of branches and $x$ represent the kernel's input. In this process, they profit from the ensemble within the block and improve the network performance. Even so, as shown in Eq. 3, the multiple outputs of the convolution layers are merged into one, which becomes a single input feature for the next block. Owing to this matter, they fail to fully exploit the ensemble effect. In other words, the sub-network topology only exists in the block illustrated in Figure. 1a, the ensemble effect does not reach the entire network. The number of sub-network topology generated by the block-level connectivity is counted as:

$$\text{\# topology with block-level connectivity} : N \times B,$$

where $N$ and $B$ represent the number of branches and blocks.

## 3.2 BRANCH-LEVEL CONNECTIVITY

The branch-level connectivity addressed in this paper is to connect the multi-branch outputs in a block to individual branches of the following block without adding or concatenating them, as shown in Figure. 1b. Therefore, unlike Eq. 3, the inputs to each convolution layers are independent as follows:

$$\mathbb{Y} = \{\mathcal{F}(x, \theta) \mid \forall \theta \in \Theta, \forall x \in \mathbb{X}\}.$$

Note that the input is now the set $\mathbb{X}$, not a single vector $x$. This ensures that the sub-network topologies are present throughout the entire network, and hence, the number of topology cases is denoted as:

$$\text{\# topology with branch-level connectivity} : N^B. \tag{4}$$

This permits neural networks to exploit the effectiveness of the ensemble with a much larger topology than block-level connectivity, but it has the critical issue of exponentially growing computational budget. We explain this matter in more detail in Sec. 4.1 regrading the computational cost.

## 4 METHOD

### 4.1 NEURAL SUBSTITUTION

Because of the ensemble effect, branch-level connectivity diversifies the topology of the sub-network, which enhances the performance even when the network is reduced by re-parameterization. However, there is a clear downside, as discussed in the view of an ensemble: the memory and computational complexity required becomes prohibitive, making practical use unfeasible. To address this downside, we present the neural substitution method

as shown in Algorithm. 1 and Figure. 3. When applying the naive branch-level connectivity, the number of input feature maps ($N$) and the number of multi-branch blocks ($N$) result in $N^2$ outputs, as indicated in Line 7 of Algorithm. 1. In contrast, the use of neural substitution method produces only $N$ output features, as shown in Line 11 of Algorithm. 1. There exists a significant difference between the naive and our neural substitution method in terms of the number of input and output features. The former naive approach exhibits exponential growth in relation to the number of blocks, while the latter ensures a consistent number of features equivalent to the inputs. Consequently, neural substitution method guarantees a fixed number of N outputs in every network, irrespective of the number of blocks involved. In addition, we incorporate the utilization of stochastic neural substitution to generate a new topology at each iteration, as seen in Lines 12-14 of Algorithm. 1. This stochastic method enables the training of a network to accommodate unlimited $N^B$ topologies as Eq. 4 using manageable computational cost with only $N$ input and output features.

---

**Algorithm 1** Pseudo code for the proposed neural substitution method. We compare the proposed branch-level neural substitution with the stochastic method with other connectivities.

---

      **Input** $X = \{x_1, x_2, ..., x_N\}$: multi-branch input features
      **Input** $\mathcal{F} = \{f_1, f_2, ..., f_N\}$: multi-branch convolution layers in a block

1:  **procedure** MULTI-BRANCH BLOCK($X, \mathcal{F}$)
2:      **switch** Connectivity **do**
3:         **case** Block-level
4:             $y \leftarrow f_1(x_1) + f_2(x_1) + ... + f_N(x_1) + ... + f_N(x_N)$
5:           **return** $y$                                           ▷ Single output
6:         **case** Branch-level (Naive approach)
7:           **return** $f_1(x_1) + f_2(x_1) + ... + f_N(x_1) + ... + f_N(x_N)$      ▷ #$N^2$ outputs
8:         **case** Branch-level (Neural substitution **w/o** stoch)
9:           **for** $i \leftarrow 0$ to $N$ **do**
10:              $y_i \leftarrow f_1(x_i) + f_2(x_i) + ... + f_N(x_i)$             ▷ Substitution
11:          **return** $y_1, y_2, ..., y_N$                         ▷ #$N$ outputs
12:         **case** Branch-level (Neural substitution **/w** stoch)
13:           $\mu \leftarrow \begin{bmatrix} 1 & ... & 1 \\ \vdots & \ddots & \vdots \\ N & ... & N \end{bmatrix} \in \mathcal{R}^{N \times N}, \quad \nu \leftarrow \begin{bmatrix} 1 & ... & 1 \\ \vdots & \ddots & \vdots \\ N & ... & N \end{bmatrix} \in \mathcal{R}^{N \times N}$
14:           $\mu \leftarrow$ shuffle $(\mu), \quad \nu \leftarrow$ shuffle $(\nu)$    ▷ Shuffle elements of index mapping matrix
15:           **for** $i \leftarrow 0$ to $N$ **do**
16:              $y_i \leftarrow f_{\mu(i,1)}(x_{\nu(i,1)}) + f_{\mu(i,2)}(x_{\nu(i,2)}) + ... + f_{\mu(i,N)}(x_{\nu(i,N)})$   ▷ Substitution
17:          **return** $y_1, y_2, ..., y_N$                         ▷ #$N$ outputs

---

## 4.2 GUIDED ACTIVATION FUNCTION

Despite successfully reducing the computing budget through our neural substitution, we are still confronted with the issue that the non-linear activation function cannot be infused through re-parameterization as proved in Eq. 2. Existing re-parameterization approaches have not been able to make use of branch-level connectivity because of this same issue. In RepVGG (Ding et al., 2021c), the first re-parameterization method to deep neural networks, they reduce the network from multiple branches into a single one, using a multi-branch architecture consisting of convolution and batch-normalization layers at only block-level connectivity. Other SOTA methods (Ding et al., 2019; 2021b; Zhang et al., 2021; Liu et al., 2018; Ding et al., 2021a; Huang et al., 2022) also involve a reduction strategy at inference time, *i.e.*, these networks also perform re-parameterization on a block-by-block basis except for non-linear activation functions, as shown in Figure. 4b. This is due to the fact that the non-linear function cannot be transformed using a linear transformation, which is why we propose employing a guided activation function as shown in Algorithm. 2 and Appendix A. The purpose of the proposed guided activation is to apply the non-linear activation function in the re-parameterization method. render the infeasible Eq. 2 feasible. This is because it is not able to reduce the output features of multiple non-linear layers as:

$$\sigma\left(\sum_{i=1}^{N} x_n\right) \neq \sum_{i=1}^{N} \sigma(x_n)$$

Therefore, to reduce the multiple non-linear layers using the proposed guided function, we devise a straightforward method to ensure that the features being activated are consistent. We adopt a similar approach as ReLU,

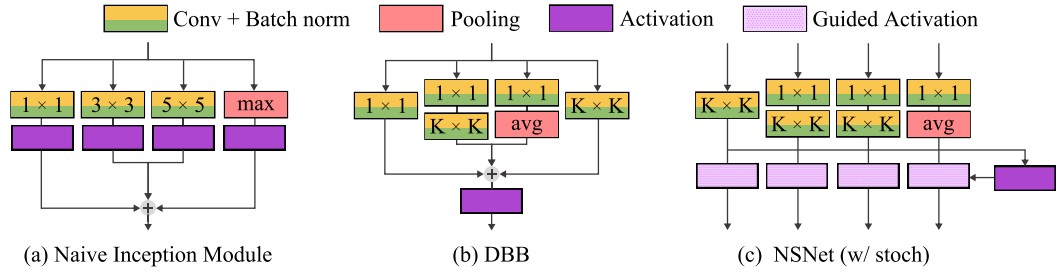

(a) Naive Inception Module     (b) DBB     (c) NSNet (w/ stoch)

Figure 4: Architectural design of re-parameterization networks. We apply our NS with the stochastic method to the DBB architecture.

which transforms all negative values of the feature to zero. We sum all multiple inputs, creating a binary guided activation mask, $Gm$, that retains only positive values, as formulated in the equation below. This mask is then applied to the preceding multiple inputs, independent of the values of the value.

$$\boldsymbol{GA}(x_i) = Gm \odot x_i; \quad Gm_{j,k} = \begin{cases} 1 & \text{if } \sum_{i=1}^{n} x_{i,j,k} > 0 \\ 0 & \text{otherwise} \end{cases}$$

$$\boldsymbol{GA}\left(\sum_{i=1}^{N} x_n\right) = \sum_{i}^{N} \boldsymbol{GA}(x_n),$$

where $x$ and $\boldsymbol{GA}$ are each feature and guided activation function. The proposed guided activation function computes the aggregated features of all branches to acquire a guided activation mask, which is applied to activate features of each branch as described in Line 2-4 of Algorithm. 2. Therefore, by employing this guided function, it is feasible to apply a non-linear activation function to multiple branch features.

---

**Algorithm 2** Pseudo code for the proposed guided activation method. This method guides the input features to be activated the same way when a network is re-parameterized.

---

**Input** $X$: input features

1: **procedure** GUIDEDACTIVATION($X = \{x_1, x_2, ..., x_N\}$)
2:     $X_s \leftarrow \sum_{i \in \{0,...,N\}} x_i$       ▷ Input feature when a network was re-parameterized
3:     $\sigma \leftarrow \mathbf{1}_{X_s>0}$                                      ▷ Guided activation mask
4:     $\tilde{X} \leftarrow X \odot \sigma$           ▷ Guiding all branch features' activation according to the mask $\sigma$
5:     **return** $\tilde{X}$

---

### 4.3 IMPLEMENTATION

The proposed neural substitution and guided activation function ameliorate existing re-parameterization methods (Ding et al., 2021c; 2019; 2021b) by applying our methods to the existing block-level connectivity as shown in Figure. 4c. To accomplish this, we do not aggregate the outputs of each branch in the branch-level re-parameterization methods; instead, we leave the branch outputs connected to the following block with the guidance of the proposed activation function. Further, the proposed method can also be used to re-parameterize multiple branches of $3 \times 3$ convolutions, where the performance boost grows as the number of branches increases. This is due to the proposed method's ability to learn a far broader range of sub-network topologies than existing block-level connectivity. The following section presents experimental results on the effectiveness of this approach of the proposed method. For a better understanding, we provide the PyTorch code in the supplementary material.

## 5 EXPERIMENT

In this section, we describe the dataset and details of our experiments in Sec. 5.1, then compare the performance of our branch-level connectivity approach to the existing re-parameterization methods in Sec. 5.2, and conduct an ablation study in Sec. 5.3 to demonstrate whether each component of our proposed method contributes to performance improvement. Finally, we introduce the analysis of the diversified representation of learned features and weight parameters due to our approach in Sec. 5.4. Note that all architectures used in the experiments, after re-parameterization, maintain the same resources such as Parameters, FLOPs, and throughput as the original backbone. Therefore, we have leave out the resource comparison.

Table 1: Experimental results of the different re-parameterization network architectures on the CIFAR100 dataset. We report Top-1 accuracy (%) for all architectures. While DBB, ACNet, ACNet$^+$ are based on the block-level connectivity, our NS method extends their connectivity to the branch-level in the re-parameterization network.

| Architecture | Reset18 | ResNet34 | ResNet50 | ResNext50 | MobileNet |
|---|---|---|---|---|---|
| Baseline | 71.90 | 73.86 | 75.52 | 75.01 | 66.42 |
| DBB | 73.70 | 74.62 | 75.56 | 76.41 | 66.69 |
| NS DBB (w/ stoch) | 74.83 | 75.09 | 76.70 | 77.25 | 66.84 |
| ACNet | 72.82 | 74.68 | 76.02 | 76.13 | 66.65 |
| NS ACNet (w/ stoch) | 73.59 | 75.38 | 76.14 | 75.90 | 66.82 |
| ACNet$^+$ | 73.53 | 74.82 | 75.81 | 76.34 | 66.55 |
| NS ACNet$^+$ (w/ stoch) | 73.61 | 74.95 | 76.06 | 76.56 | 66.77 |
| NSNet (w/ stoch) | **75.36** | **75.72** | **77.72** | **77.64** | **67.33** |

Table 2: Experimental results of block- and branch-level re-parameterization network architectures on the ImageNet dataset.

| Architecture | Reset18 | ResNet50 | MobileNet |
|---|---|---|---|
| Baseline | 69.19 | 77.31 | 70.45 |
| DBB | 70.13 | 77.50 | 70.78 |
| NS DBB (w/ stoch) | **70.75** | 77.53 | 72.06 |
| ACNet | 70.02 | 77.30 | 71.12 |
| NS ACNet (w/ stoch) | 70.41 | 77.61 | **72.39** |
| ACNet$^+$ | 70.25 | 77.48 | 71.39 |
| NS ACNet$^+$ (w/ stoch) | 70.43 | **77.72** | 72.04 |

Table 3: Experimental results of the MobileOne architecture.

| CIFAR100 | |
|---|---|
| Architecture | Accuracy |
| MobileOne | 67.26 |
| + NS (w/ stoch) | **67.48** |
| ImageNet | |
| Architecture | Accuracy |
| MobileOne | 72.30 |
| + NS (w/ stoch) | **72.76** |

## 5.1 DATASET AND SETUP

In our experiments, we employ two standard visual classification datasets: ImageNet (Deng et al., 2009) and CIFAR100 (Krizhevsky et al., 2009). In the ImageNet dataset, networks are trained with a batch size of 2048, and the learning rate is $3.5e^{-3}$, which is scheduled with a warm-up and a cosine function during 100 epochs. The LAMB optimizer is used with an EPS of $1e^{-6}$ and a weight decay of 0.02. In the CIFAR100 dataset, all networks adopt a batch size of 1024 with 100 epochs. The details of the training are provided in Appendix B.

## 5.2 EXPERIMENTAL RESULTS

We utilize ResNet (He et al., 2016), ResNext (Xie et al., 2017), and MobileNetV1 (Howard et al., 2017) as the backbone networks and replace every convolution block to multi-branched one as shown in Figure. 4b and c. To apply our approach to the existing block-level re-parameterization methods, we use five architectures such as DBB (Ding et al., 2021b), ACNet (Ding et al., 2019), ACNet$^{+1}$, and MobileOne (Vasu et al., 2023).

Table. 1, 2, and 3 demonstrates the efficacy of our neural substitution (NS) method works favorably against all the re-parameterization network architectures on both CIFAR100 and ImageNet dataset. It is evident that our NS constantly enhances the performance, regardless of the specificities of the re-parameterization network design and backbones. Furthermore, using our architecture, Neural Substitution Networks (NSNet), we improve the performance of re-parameterization method significantly compared to existing architectures.

## 5.3 ABLATION STUDY

**Branch-level connectivity:** Table. 4 shows the observed performance improvement according to the number of branches. Multiple $3 \times 3$ convolutions are employed for all multi-branch designs. Once the number of $3 \times 3$ convolutions surpasses four, the block-level connection experiences performance saturation. However, our

---

[1]This architecture use one more $1 \times 1$ convolution branch in a block to ACNet.

Table 4: Experimental comparison between block- and branch-level connectivities regarding the number branches.

| # branch | Block-level | Branch-level NS |
|---|---|---|
| Conv×2 | 73.16 | 73.30 (+0.14) |
| Conv×3 | 73.53 | 74.10 (+0.57) |
| Conv×4 | 74.15 | 74.58 (+0.43) |
| Conv×5 | 74.20 | 74.79 (+0.59) |

Table 5: Experimental results for the ablation study of the proposed neural substitution method with the stochastic method.

| Architecture | DBB | ACNet | ACNet$^+$ |
|---|---|---|---|
| Block-level | 70.13 | 70.02 | 70.25 |
| NS w/o stoch | 70.56 | 70.21 | 70.38 |
| NS w/ stoch | 70.75 | 70.41 | 70.43 |

Table 6: Transfer learning accuracy with regard to NS MLP and classifier on 9 datasets (Krizhevsky et al., 2009; Thomee et al., 2016; Cimpoi et al., 2014; Maji et al., 2013; Bossard et al., 2014; Parkhi et al., 2012; Cao et al., 2021; Krause et al., 2013; Deng et al., 2009). To leverage the characteristics of NS that requires the creation of multiple topologies, we opt for the use of multiple layers of MLP as visual adapters instead of a using only single classifier. Archi, C100, Co211, INet, and Avg respectively refer to architecture, CIFAR100, Country211, imageNet, and average. The three architectures have learnable parameters of sizes 0.05M, 5M, and 5M in order. The FC and MLP denotes the classifier and vision adapter.

| Archi. | C100 | Co211 | DTD | FGVC | Food | Pet | Pcam | Cars | INet | Avg. |
|---|---|---|---|---|---|---|---|---|---|---|
| FC | 77.12 | **26.09** | 70.48 | 42.00 | 88.20 | 88.77 | 78.37 | 80.36 | 75.08 | 69.61 |
| MLP/FC | 78.29 | 23.56 | **71.65** | 48.55 | 88.27 | **90.92** | 67.17 | **82.47** | 76.55 | 69.71 |
| NS MLP/FC | **79.77** | 24.74 | 70.59 | **50.86** | **88.86** | 90.57 | **83.37** | 82.34 | **76.81** | **71.99** |

branch-level NS architecture consistently demonstrates improved performance over the block-level architectures in all cases, including the five convolution branches.

**Stochastic method of neural substitution:** Table. 5 is the experimental result of the stochastic method, which randomizes the substitution method at every iteration. According to this result, the proposed stochastic method is beneficial in all cases. We observe that employing the proposed NS with our stochastic method improves the network's performance by $0.18 \sim 0.62\%$ compared to the block-level connectivity.

**Neural substitution on MLP** In order to show the effectiveness of the proposed neural substitution method in the Vision Transformer architecture, we perform further experiments using the ViT-B/32 architecture of CLIP (Radford et al., 2021). As a result, we demonstrate that the proposed neural substitution method improves the performance of the original ViT-B/32 model in most cases as shown in Table. 6.

## 5.4 ANALYSIS

**Block and Branch Similarity** In order to establish the proposition that our branch-level connectivity indeed makes use of more topologies compared to the block-level one, we examine the feature and weight similarity of branches in a block. We conduct this similarity analysis to assess the feature diversity in the re-parameterization method. The more topologies a network has, the more different features with low similarity it will learn between branches within a block. This enhanced feature diversity produces low feature similarity and leads to better performance for the re-parameterized networks. In this regard, we estimate the feature similarities in Figure. 5. In Figure. 5a, dot points are cosine similarity between pixels at corresponding locations in feature maps of layers at varying depths. While similarities of block-level connectivity are scattered above 0.6, our branch-level connectivity exhibits far lower similarity values. Figure. 5b shows the feature cosine similarity between pixels of different branches in a block. As the number of branches increases, our branch-level connectivity has a lower similarity compared to the block-level one. These two similarity analyses demonstrate that our branch-level approach offers richer learned features through the unlimited network topologies in a network.

**Kernel visualization.** We visualize the kernel weight parameters of different branches in a block. As shown in Figure. 6a, block-level connectivity in the top row exhibits a consistent trend of all branches of $3 \times 3$, $1 \times 1$, $1 \times 3$, and $3 \times 1$ kernels with over smoothed weight parameters. Therefore, it shows the monotonic patterns of re-parameterized weight parameters. In contrast, the weight parameter of the proposed branch-level connectivity NS (middle and bottom rows) offers a variety of patterns. This result further supports the notion that our re-parameterization method at the branch level promotes the generation of richer learned features.

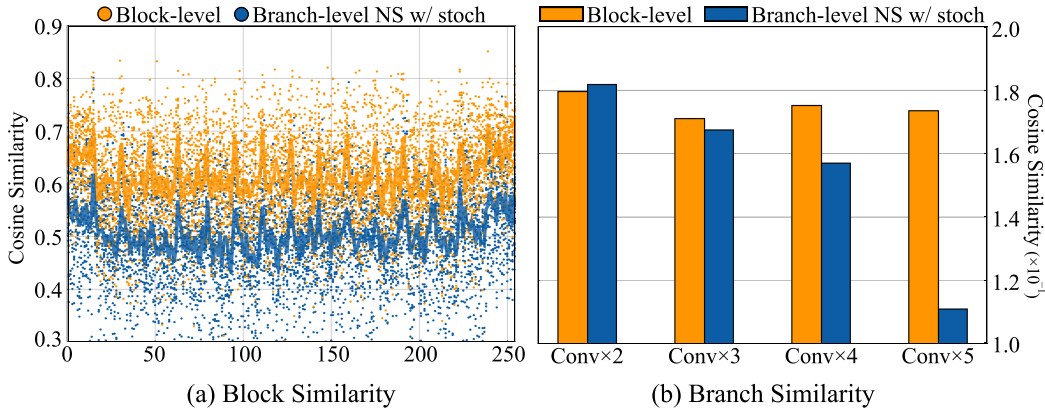

(a) Block Similarity       (b) Branch Similarity

Figure 5: Feature similarity comparison. (a) The cosine similarity is measured on the features between blocks at different depths of a network. (b) The cosine similarity between features of different branches of a block is computed.

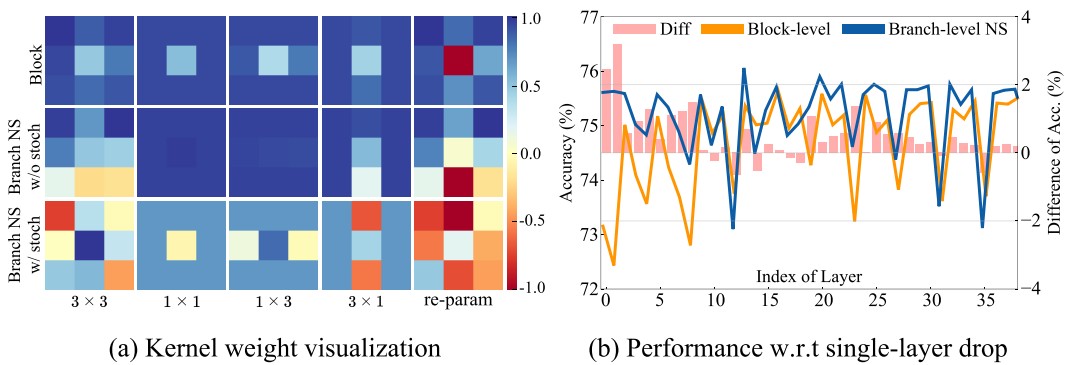

(a) Kernel weight visualization       (b) Performance w.r.t single-layer drop

Figure 6: (a) Kernel weight visualization for branches in a block. Our branch-level NS method shows more differing representations of weight parameters due to its unlimited topologies. (b) Performance degradation due to single block drop. We report the performance degradation when a single block is dropped in a network. The red bar is calculated by $Branch_{Acc} - Block_{Acc}$. Our branch-level NS is more resilient to the single layer drop.

**Ensemble effect.** As discussed in earlier study (Veit et al., 2016), the absence of a single block in a network may not significantly impair performance due to the ensemble effect (He et al., 2016). Regarding this, Figure. 6b illustrates that our NS enhances the benefits of the ensemble effect when a single block is dropped. In this illustration, it is shown that our NS consistently performs better than the branch-level connectivity, which substantiates our claim that NS improves the benefits of ensembles by generating unlimited topologies in the re-parameterization process.

## 6 CONCLUSION

In this paper, we aim to the re-parameterization with the branch-level connectivity, which enriches the sub-network topology in the training step. Unavoidably, the branch-level connectivity suffers from exponentially expanding the computation of the entire network during training and failing to infuse the multiple non-linear activation function. To overcome this challenge, we propose the stochastic neural substitution and guided activation function. To reduce the computational budget, the proposed method stochastically decides the substitution in each iteration and uses the guided function to linearly transform the non-linear activation function. Our experiments reveal that the proposed method outperforms existing block-level connectivity, and an ablation study demonstrates the usefulness of our strategy. The proposed method, we believe, will be useful in future research direction.

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

## A    EXAMPLE OF GUIDED ACTIVATION

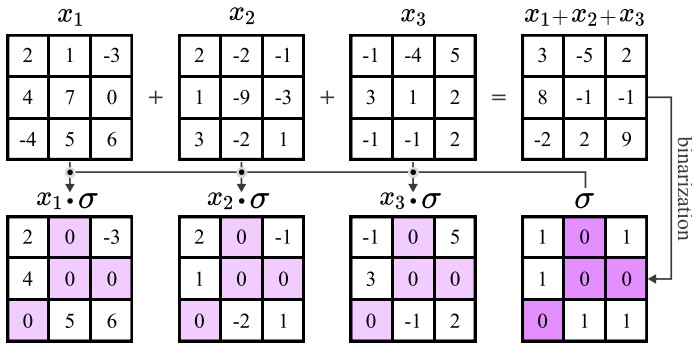

Figure 7: Workflow of the proposed guided activation function. $x_i$ denotes the feature map of each branch and $\sigma$ is the guiding activation map. The $\sigma$ guides the activation of $x_i$.

## B    DETAILS OF HYPER-PARAMETER SETTING

Table 7: Details of the hyperparameter settings for the ImageNet and CIFAR100 datasets. Note that in CIFAR100 and imageNet, MobileNetV1 and MobileOne utilize 128 and 512 batch sizes. The VLM means that experiment setting in Table. 6 with 9 datasets.

| Dataset | CIFAR100 | ImageNet | VLM |
|---|---|---|---|
| Epochs | 100 | 100 | 30 |
| Batch size | 2048 | 1024 | 256 |
| Optimizer | LAMB | LAMB | AdamW |
| Weight decay | $1.0e^{-2}$ | $1.0e^{-2}$ | $1.0e^{-2}$ |
| LR(Learning rate) | $3.5e^{-3}$ | $3.5e^{-3}$ | $3.5e^{-3}$ |
| Warmup epoch | 5 | 3 | 5 |
| Warmup LR | $1.0e^{-5}$ | $1.0e^{-4}$ | $1.0e^{-5}$ |
| Min LR | $1.0e^{-6}$ | $1.0e^{-6}$ | $1.0e^{-5}$ |
| Image size | $3 \times 32 \times 32$ | $3 \times 224 \times 224$ | $3 \times 224 \times 224$ |
| Label smoothing | 0.1 | 0.0 | 0.0 |
| Rand Augment | X | 7 / 0.5 | 7 / 0.5 |
| Auto Augment | CIFAR10 policy | X | X |
| Cutmix | 0.0 | 1 | 0.0 |
| Mixup | 0.0 | 0.1 | 0.0 |
| Loss | Cross Entropy | Binary Cross Entropy (0.2) | Cross Entropy |
| Color Jitter | 0.0 | 0.4 | 0.0 |
| Train interpolation | bicubic | random | bicubic |
| Test interpolation | bicubic | bicubic | bicubic |
| Test crop ratio | 1.0 | 0.95 | 1.0 |
| Stoch. Depth | 0.15 | 0.05 | 0.0 |

