# OpenReview forum: "Branch-level Network Re-parameterization with Neural Substitution"
_ICLR.cc/2024/Conference — Submitted to ICLR 2024_

### Official Review · Reviewer_NvSB · 2023-11-02

**Soundness:** 2 fair
**Presentation:** 2 fair
**Contribution:** 2 fair
**Rating:** 3
**Confidence:** 3

**Summary:**

This paper studies the problem of better strategies for network reparameterization. This general line of work involves reparameterizing a potentially more complicated train time architecture into a specific test time architecture. The goal is to simplify and preserve performance, and different approaches make different design choices regarding the level of granularity at which the parameterization will be carried out. This work proposes reparameterizing at the branch level and a neural substitution scheme is presented. But this low-level of granularity creates a problem with handling the activation function, and therefore, a guided activation scheme is presented. Experiments show nominal gains.

**Strengths:**

1. Explores branch level re-parameterization which appears to have not been attempted in the literature.

2. Guided activation applies reparameterization on the activations.

3. Some gains in experiments are achievable.

**Weaknesses:**

1. The paper is not well written. The high level goal of this work should be laid out clearly but instead the paper assumes that the reader is intricately familiar with most of the existing works on re-parameterization.

2. While I appreciate the interest in branch-level reparameterization, eventually the paper must be able to make a convincing case that the effort is worth it. The gains are marginal at best and don't appear to match up with other works (e.g., RepVGG). The choice of baselines is not explained.

3. The technical contribution (neural substitution, guided activation) are somewhat minor. By themselves, this would not be a weakness if the experiments were comprehensive enough to show their value. Latency, memory footprint and other relevant issues that are central discussion points in other papers on this topic appear to be omitted.

**Questions:**

I find the general goal of this work relevant but find the degree of novelty in the main contributions and the overall rigor/comprehensiveness of the experiments (e.g., compared to RepVGG) underwhelming. Please explain if possible why these concerns are misplaced or incorrect.

---

> ### Author Response · Authors · 2023-11-23
> **Rebuttal: baseline selection and resource usage.**
>
> Following the reviewer’s comments, we have included more explanations to improve the readability in understanding the proposed re-parameterization method. We thank you for the constructive comment.
>
> ***RepVGG***: ACNet and DBB are the improved version of RepVGG model. Therefore, we show that ours performs better than ACNet and DBB, and thus we can guess that ours performs well compared to RepVGG. RepVGG mentioned that using the activation function in the re-parameterization method can improve performance, but they were not able to use the non-linear activation function in their method. However, we propose a re-parameterization method based on branch-level connectivity with the non-linear activation function, resulting in further improved performance compared to block-level connectivity based re-parameterization methods.
>
> ***Resource usage***: Since the re-parameterization methods merge the branched convolutional layers into a single one after training is completed, they use the same network architecture as the original backbone for inference. Therefore, all re-parameterization methods use the same latency and memory footprint as the original backbone.

---

### Official Review · Reviewer_LE6H · 2023-11-03

**Soundness:** 3 good
**Presentation:** 3 good
**Contribution:** 3 good
**Rating:** 6
**Confidence:** 3

**Summary:**

This is the comments from the fast reviewer.

In this study, the authors introduce a novel network reparameterization approach named
"Neural Substitution," which utilizes an unprecedented range of network topologies to
achieve more complex representations than current methodologies allow. By converting the
nonlinear activation function into a linear transformation, this method successfully navigates
past the constraints of traditional reparameterization techniques. The authors employ a
process they term "Guided Activation," replacing the nonlinear activation with a linear
counterpart to simplify computational demands while preserving the benefits inherent in a
multi-branch architecture. Validation across various datasets and network configurations
affirms the superiority of Neural Substitution over current block-level connectivity
approaches, a claim further substantiated by the results of an ablation study. Comparisons
with other reparameterization strategies highlight the proposed method's enhancements in
computational efficiency and model performance. Overall, Neural Substitution emerges as a
promising reparameterization method, poised to significantly impact future explorations in the
field.

**Strengths:**

as it claimed

**Weaknesses:**

The paper does encounter some challenges in terms of readability, with certain sections
displaying a propensity for overly elongated sentences that may impede comprehension and
introduce potential for misinterpretation. For instance, the introduction's discussion of
contributions is excessively verbose, obscuring the method's branch-level connectivity
advantages. Additionally, the background segment on structural reparameterization attempts
to condense an excessive amount of information into a single sentence.

Regarding the presentation of methods, the Neural Substitution section is eloquently
articulated, enlightening us to a sequence shuffle-based technique for diminishing the
dimensions of branch-level connectivity. However, the explanation within the Guided
Activation Function segment lacks specificity. It presents the principles of the Guided
Activation Function solely through an algorithmic lens without a direct comparison to
traditional methods, leaving the reader puzzled about its actual impact and application.

**Questions:**

- How does the proposed method compare to existing methods in terms of computational efficiency?
- Can the proposed method be applied to other types of neural networks, such as recurrent neural networks？
- How does the proposed method handle overfitting, and what measures were taken to prevent it during the experiments?

---

> ### Author Response · Authors · 2023-11-23
> **Rebuttal: computational efficiency, applying to RNN and overfitting.**
>
> Our work has been updated to better convey our points and make it easier to read based on the reviewer’s comments. We appreciate the helpful comments from the reviewer. We shorten the sentences in the introduction to make it easier to read. Additionally, we revised our paper to give a comprehensive explanation for the proposed method in `Sections 1 and 2`. In addition, we thought that our explanation of guided activation was too brief, so we added more equations to clarify the necessity of guided activation. We updated equation in `Section 4.2`  to make it easier for readers to understand.
>
> ***Computational cost***: All the re-parameterization networks use multiple convolution layers in the training stage, but they merge the multiple convolution layers into a single one. Therefore, the computational cost of the baseline original network and re-parameterization networks is the equivalent.
>
> ***RNN result***: RNNs can be re-parameterized analytically as well. We constructed a non-linearly connected parallel fully connected layer and implemented it on the visual adapter of CLIP, the vision language model. Our neural substitution shows a significant improvement in performance in this experiment, `Table. 6` illustrates this.
>
> ***Overfitting***: As a means of addressing overfitting, we applied stochastic substitution during the training phase of neural substitution. As a result, the model can learn more generalized features.

---

### Official Review · Reviewer_AsUr · 2023-11-04

**Soundness:** 2 fair
**Presentation:** 2 fair
**Contribution:** 2 fair
**Rating:** 5
**Confidence:** 3

**Summary:**

This paper introduces a branch-level re-parameterization method for transforming a residual-based model into a plain model. The paper leverages two primary techniques: neural substitution and guided activation. In particular, the neural substitution method enhances the ensemble effect of convolutional models, while the guided activation method facilitates the reparameterization of the non-linear activation function within the block. Experimental validation is performed on the CIFAR-100 and ImageNet datasets.

**Strengths:**

- This paper delves into the realm of designing plain models through re-parameterization literature. The proposed pipeline optimizes the utilization of ensemble effects in deep neural network models while circumventing the limitation of previous re-parameterization techniques, where non-linear activation had to be located outside the block. This research direction is intriguing and offers a valuable complement to existing re-parameterization methods.

**Weaknesses:**

- Towards method
    - Regarding Neural Substitution, can you clarify whether you perform random shuffling of $\mu$ and $\nu$ during batch training, as mentioned in lines 12 to 14 of Algorithm 1? If so, could you provide insights on how to apply $\mu$ and $\nu$ during inference?
    - When it comes to Guided Activation, what is the rationale behind using the sum of features as a guiding factor for activation? Could you provide an explanation for this design choice? It appears that the guided activation (as shown in Figure 5) may not be formally equivalent to the original CNN structure, like ResNet, where non-linear activation resides within the block.

- Towards experiments
    - Merely reporting accuracy is not sufficient to demonstrate the superiority of your method. Considering that accuracy and speed can often be a trade-off in network design, it is advisable to include additional metrics, such as speed, parameters, and FLOPs (in reference to RepVGG [R1]).
    - Even when focusing on accuracy alone, the current evidence may not be adequate to establish the superiority of your method. In many settings, the improvement appears to be marginal, often less than 1%.
    - Figure 6 appears to be empty.

- Minor: There should always be a space between a word and its following reference. For example, 'The method RepVGG (Ding et al., 2021c)' instead of 'The method RepVGG(Ding et al., 2021c).'

[R1] RepVGG: Making VGG-style ConvNets Great Again

**Questions:**

See *Weaknessnes*

---

> ### Author Response · Authors · 2023-11-23
> **Rebuttal: Shuffle in inference time and performance.**
>
> Our paper has been strengthened with your constructive comments. Reviewer’s comments have been reflected, and `Figure. 5` is now updated for easy viewing.
>
> ***Shuffle operation***: In `Figure. 3`, a kernel that uses b1 as input would never encounter $a_4$ to $a_9$ during training if shuffle wasn't used, and $b_1$ would always equal $a_1+a_2+a_3$. Therefore, less diversified topologies will be used in the training stage due to this fixed connection. To make diverse connections, we use shuffle operation so that $b_1$ can use $a_1$ through $a_9$ during the training stage. Following training, we integrate all convolution layers into a single layer as illustrated in `Figure. 2`. Therefore, we do not use the shuffle operation for inference. Moreover, the line 12~14 of `Algorithm. 1` produces the index mapping matrix to shuffle input features.
>
> ***Resource specification***: All networks use identical resources, such as FLOPs, parameters, and throughput, after re-parameterization. In the similar vein of the shuffle operation, we merge all branched convolution layers into a single layer after re-parameterization, so all the re-parameterization networks exploit the same resources.
>
> ***Performance***: In the rebuttal phase, we constructed our own architecture to show further improved performance in `Table.1`. Therefore, it is shown that the performance is much better than the submitted version of the paper and performs favorably against existing methods.
>
> We have also applied our method to Vision Transformer and found that the performance is also improved in `Table. 6`.

---

### Official Review · Reviewer_Gpid · 2023-11-04

**Soundness:** 2 fair
**Presentation:** 1 poor
**Contribution:** 2 fair
**Rating:** 3
**Confidence:** 2

**Summary:**

The paper is focused on branch-level network re-parameterization using neural substitution. The authors propose a generalized framework that can be applied to various network designs with branch-level connectivity and re-parameterizable non-linear activation functions. They demonstrate that their proposed method outperforms existing re-parameterization methods and can be adapted to different architectures.

**Strengths:**

The paper introduces a approach called Neural Substitution (NS) for network re-parameterization at the branch-level connectivity. This approach addresses the computational complexity issues of heavy multi-branch designs by substituting local paths. The authors also propose the use of guided activation methods to reduce non-linear activation functions during the re-parameterization process.

**Weaknesses:**

-  The paper would benefit from enhancing its clarity and readability. It currently lacks a "Related Work" section, which is essential to contextualize the research and provide a comprehensive background.

- The content, particularly mathematical expressions, requires further clarification. For instance, in equation (1), the inclusion of a bias term $\beta$ in $\theta$ may be necessary for completeness. Additionally, there seems to be a typographical error in the subsequent equations where $\theta_1$ should be replaced with $\theta_2$. The rationale behind reducing only the first two $\theta$'s into $\widetilde\theta$ is not clear to me and needs explanation.

- The main idea is rather simple, more importantly, advancements over existing methods seem marginal, with many improvements being within a 0.3% accuracy range, which could be considered insignificant.

- The applicability of the proposed technique is demonstrated solely on CNN architectures. Its effectiveness on other architectural designs, such as fully connected layers or Transformers, is not addressed, raising concerns about its generalizability.

**Questions:**

See the weaknesses part above.

---

> ### Author Response · Authors · 2023-11-23
> **Rebuttal: performance and applying to the architecture other than CNN.**
>
> We have polished our paper to improve readability and to make our claims clearer. We appreciate the reviewer for the precious comments. Specifically, we have added more detailed explanations in `Section. 2`. Related work and `Figure. 4` to make it easier for the reader. We have also updated the formulas for the clear understanding.
>
> ***Performance***: For both Cifar and ImageNet dataset, we show that our proposed method consistently 1) **outperforms existing network re-parameterization methods**, and 2) **provides significant improvement over the original baseline network**. Further, in the rebuttal phase, we introduce new architecture for our re-parameterization method. Thus, we show that ours further improve the performance of Cifar dataset. While we didn't confirm the performance improvements on ImageNet dataset due to the limited time of the rebuttal phase. However, we are confident that we can improve performance on the ImageNet dataset and will continue to experiment after the rebuttal phase.
>
> ***Vision Transformer***: We apply the proposed branch-level re-parameterization method for the Vision Transformer on the vision language model. We confirm that ours improves the performance of Vision Transformer in `Table. 6`.

---

### Official Review · Reviewer_j2y2 · 2023-11-06

**Soundness:** 2 fair
**Presentation:** 2 fair
**Contribution:** 2 fair
**Rating:** 3
**Confidence:** 5

**Summary:**

The paper propose a branch-level network re-parameterization method,  including two core design, neural substitution and guided activation.  Neural substitution is used to aggregate features from multiple branches (the number is N). Unlike traditional block-level methods which produces single output, it generates N outputs for feeding into next blocks,  encouraging the learning of richer representations.  The guided activation utilizes the activation mask of the aggregated features of all branches to threshold the generated outputs. Experiments on multiple backbone architectures and classification datasets show that the proposed method improves representation diversity across branches and can obviously improve the accuracy compared to baseline methods.

**Strengths:**

1. The work aims to enhance the richness and diversity of multi-branch networks, which is a valuable direction of research. The proposed strategy is easy to implement and can be readily integrated into some widely-used backbone networks.

2. The authors provide pseudo code and illumination for the algorithm, which is helpful for understanding the work.

3. Experimental results show that using the method can obviously benefit the accuracy of baseline networks (e.g., DBB and ACNet). The authors provide analysis for its advantage on feature diversity across branches and ensemble effect when dropping a single block.

**Weaknesses:**

1. The design and implementation of the work is build on block-level re-parameterization architectures, while the core of the work seems less matching the motivation of network re-parameterization. Network re-parameterization works mainly target at decoupling the training-time and inference-time network structure, typically complicating the training models but converting it into a simple architecture for inference by the consideration of deployment and efficiency. The operation neural substitution introduces  Nx memory overhead on feature aggregation compared to block-wise methods (even more for catching the intermediate input).  The method actually addresses the feature diversity of multi-branch, through a better aggregation strategy, rather than network re-parameterization.

2. The authors need to provide the analysis on memory overhead, which is an important metric for real-world application.

3. The description for the method need to be improved. The section of method and introduction lack of sufficient description for the mechanism and the insight behind, although algorithms (1 and 2) and illumination (figure 3 and 5) are provided to explain the method.  For example, I think the stochastic operation (shuffle) can largely improve the richness of features compared to without it. The authors emphasize multiple times about ``nearly unlimited topology of branch-level connection’’, if using the setting without stochastic, aggregating features in a regular order is intractable to approach it. Moreover, the guided activation is designed specially for ReLU nonlinearity, which needs to be discussed.

4. It would be valuable to see the generalization on different CV tasks in experiments beyond classification.

**Questions:**

1. The figure 6 shows the feature similarity comparison. It would be interesting to see if the similarity reduction mainly comes from stochastic or neural substitution.

2. I notices some differences between the performance of baseline methods in the paper and the ones reported in the DBB paper. For example, in DBB paper, the results on ImageNet with MobileNet, ResNet-18 and ResNet50 are (72.88, 70.99, 76.71) for DBB-Net and (72.14, 70.53, 76.46) for ACNet. In Table 2, the corresponding results are (70.78, 70.13, 77.50) for DBB-Net and (71.12, 70.02, 77.30) for ACNet. Some of the values are higher but the others are lower.  I think the gap may derive from the configuration difference. It would be highly recommended to report the results according to the configuration of baseline method (e.g., DBB) as well, enabling a more fair comparison.

---

> ### Author Response · Authors · 2023-11-23
> **Rebuttal: resource overhead, stochastic substition, performance of existing networks.**
>
> We appreciate that the reviewer’s valuable comments have improved our paper. Below are answers to the reviewer's concerns.
>
> ***Resource overhead***: Although there is a memory overhead during training, re-parameterization allows for inference to have the same memory, FLOPs, and throughput as the original backbone after training. It is evident from other re-parameterization networks, such as ACNet, DBB, and RepVGG, use the same parameters, FLOPs, and throughput. Because of this, we did not report memory overhead in this work.
>
> ***Unlimited topology, richness of features***: The convolution procedure will result in a total of nine features which we named features $a_1,a_2,\ldots,a_9$ and convolution $F_1, F_2$, and $F_3$ from three inputs and three convolutions, as illustrated in Figure. 3. In the absence of stochastic training, $F_1$ will only ever exploit $a_1, a_2$, and $a_3$ by the substitution; it will never use $a_4,a_5,\ldots,a_9$, which means that the topology won't be enriched. Since we want F_1 to learn every feature of  $a_1,a_2,\ldots,a_9$ , stochastically switching $a_1,a_2,\ldots,a_9$  in each training iteration would allow us to train nearly infinite topologies.
>
> ***Vision Transformer***: We have further demonstrated that the proposed method improves the performance of the Vision Transformer in `Table. 6`, using the vision language model.
>
> ***Performance of existing networks***: The reason why the performance of existing re-parameterization networks such as DBB is different from the original paper is that we performed our experiments in the same experimental environment by ourselves.
>
> Each existing network has a slightly different experimental environment, such as hyperparameters, and we made them all the same so that we could perform all the experiments and report the results for a fair comparison. In our experiments, the performance of the existing networks of our experiments closely resembles that of the original study, as does the overall trend.

---

### Author Response · Authors · 2023-11-23
**Additional experiments using a novel architecture and verification in the context of ViT.**

Through the constructive reviews of the reviewers, we have enhanced our manuscript to present our contributions in a more coherent and refined manner. We have improved the explanation and mathematical equations of the proposed method.

We summarize our response to the comments raised by reviewers.

***Resources*** (Parameters and FLOPs): We clarify that when we deploy the model after training with the re-parameterization, all of its resource specifications (such as throughput, parameters, and FLOPs, etc.) are the same as the original backbone. Put otherwise, the parameters, FLOPs, and throughput of original ResNet50 and the ResNet50+re-parameterizable architecture design with Neural Substitution are identical. This is because the branched multiple layers are merged into a single layer by re-parameterization after training, so they are the same as the original backbone. To make it clear, we revised `Section. 5` to include the claim that **the model specification remains unchanged at inference time.**

***Performance***: During the revision period, we designed a new network architecture that can better learn our branch-level reparameterization. As shown in `Tables 1`(below table), it is shown that the performance with our new architecture is considerably better compared to other methods. The updated text of the proposed architecture can be found in `Figure. 4c` of the revised manuscript.

|     Architecture     |  Reset18   |  ResNet34  |  ResNet50  | ResNext50  | MobileNet  |
| :------------------: | :--------: | :--------: | :--------: | :--------: | :--------: |
|       Baseline       |   71\.90   |   73\.86   |   75\.52   |   75\.01   |   66\.42   |
|         DBB          |   73\.70   |   74\.62   |   75\.56   |   76\.41   |   66\.69   |
|  NS DBB (w/ stoch)   |   74\.83   |   75\.09   |   76\.70   |   77\.25   |   66\.84   |
|        ACNet         |   72\.82   |   74\.68   |   76\.02   |   76\.13   |   66\.65   |
| NS ACNet (w/ stoch)  |   73\.59   |   75\.38   |   76\.14   |   75\.90   |   66\.82   |
|        ACNet+        |   73\.53   |   74\.82   |   75\.81   |   76\.34   |   66\.55   |
| NS ACNet+ (w/ stoch) |   73\.61   |   74\.95   |   76\.06   |   76\.56   |   66\.77   |
|   NSNet (w/ stoch)   | **75\.36** | **75\.72** | **77\.72** | **77\.64** | **67\.33** |

***Vision Transformer***: To verify that the proposed method also works on Vision Transformer, we performed an evaluation on the vision language model CLIP [1]. As shown in `Table. 6`(below table), it is confirmed that our method improves the performance of Vision Transformer.

|  Architecture   |  Cifar100  | Country211 |    DTD     |    FGVC    |  Food101   |    Pet     |    Pcam    |    Cars    | imageNet   | Avg       |
| :-------------: | :--------: | :--------: | :--------: | :--------: | :--------: | :--------: | :--------: | :--------: | :----------: | :---------: |
| FC (classifier) |   77\.12   | **26\.09** |   70\.48   |   42\.00   |   88\.20   |   88\.77   |   78\.37   |   80\.36   | 75\.08     | 69.61     |
|     MLP/FC      |   78\.29   |   23\.56   | **71\.65** |   48\.55   |   88\.27   | **90\.92** |   67\.17   | **82\.47** | 76\.55     | 69.71     |
|    NS MLP/FC    | **79\.77** |   24\.74   |   70\.59   | **50\.86** | **88\.86** |   90\.57   | **83\.37** |   82\.34   | **76\.81** | **71.99** |

-----

[1] Radford, Alec, et al. "Learning transferable visual models from natural language supervision." International conference on machine learning. 2021.

---

### Meta-Review · Area_Chair_xD1c · 2023-12-08

**Metareview:**

This paper presents a method for re-parameterizing network branches, which is composed of two key designs: neural substitution and guided activation.

Neural substitution is used to aggregate features from multiple branches. Unlike traditional block-level methods that produce a single output, it generates multiple outputs for feeding into the next blocks, promoting the learning of richer representations.

Guided activation, on the other hand, uses the activation mask of the aggregated features from all branches to threshold the generated outputs.

The experiments conducted on various backbone architectures and classification datasets (Cifar100 and ImageNet) demonstrate that this method enhances representation diversity across branches and significantly improves accuracy compared to baseline methods.

The reviewers have various concerns on the presentation quality, contribution and novelty of the methodology, as well as experimental details. Majority of the reviewers suggest a rejection with one slightly above the borderline. So a rejection is suggested.

**Justification For Why Not Higher Score:**

Majority of the reviewers suggest a rejection with one slightly above the borderline. So rejection is suggested.

**Justification For Why Not Lower Score:**

N.A.

---

### Decision · Program_Chairs · 2024-01-16

Reject